# Shifting Models Left: End-to-End Traceability as a Foundation for Knowledge Graphs as Data Products

Lionel Tailhardat[1,*], Joanna Balcerzak[1], Arij Elmajed[1,*], Pano Maria[2,*] and Gabriel Fomi-NGameni[1]

[1]*Orange Research, France*

[2]*ModelDesk, The Netherlands*

## Abstract

Various methods and tools already exist in the field of knowledge graph construction, which can be combined into pipelines for knowledge graph-based decision support systems. However, the semantic integration of multiple data sources into a knowledge graph is a craftsmanship task closely linked to the knowledge/data engineers' deep understanding of the information system they work in. This vision paper proposes to approach the problem from the opposite side of the pipeline, rather than assembling integration logic bottom-up, we start from the hypothesis of end-to-end traceability, the verification mechanism that makes "shifting semantics left" operational, and views the knowledge graph as a data product. Our proposals are grounded in the logical nature of Semantic Web technologies and the concept of formal proof of systems.

## Keywords

Semantic integration, Data lineage, Schema matching, Data mesh, RML, SPARQL, Formal proof

## 1. Introduction

Declarative approaches for Knowledge Graph Construction (KGC), such as the RDF Mapping Language (RML) [1] and related tools, have seen significant growth in recent years due to the enthusiastic involvement of scientific, technical, and industrial communities around having a common and logical framework to describe and implement data integration systems. Research in diverse related fields such as incremental data integration [2], ontology modifications propagation [3], the use of LLMs to produce KGC pipelines [4], and the cartography of Semantic Web-related tools [5] attests to this progress. This body of work forms a corpus of methods and tools that address fundamental needs for reliable semantic integration of heterogeneous data, thereby bringing the benefits of knowledge graphs to real-world knowledge bases and decision support systems [6, 7, 8].

Within this body of work, reflections on end-to-end and industry-grade KGC pipelines

*KGCW'26: 7th International Workshop on Knowledge Graph Construction, May 10, 2026, Dubrovnik, Croatia*

*Corresponding author.

✉ lionel.tailhardat@orange.com (L. Tailhardat); joanna.balcerzak@orange.com (J. Balcerzak); arij.elmajed@orange.com (A. Elmajed); pano@modeldesk.io (P. Maria); gabriel.fomingameni@orange.com (G. Fomi-NGameni)

🆔 0000-0001-5887-899X (L. Tailhardat); 0009-0007-0396-1412 (J. Balcerzak); 0009-0007-2969-9141 (A. Elmajed); 0009-0000-2598-1894 (P. Maria); 0009-0004-7908-2463 (G. Fomi-NGameni)

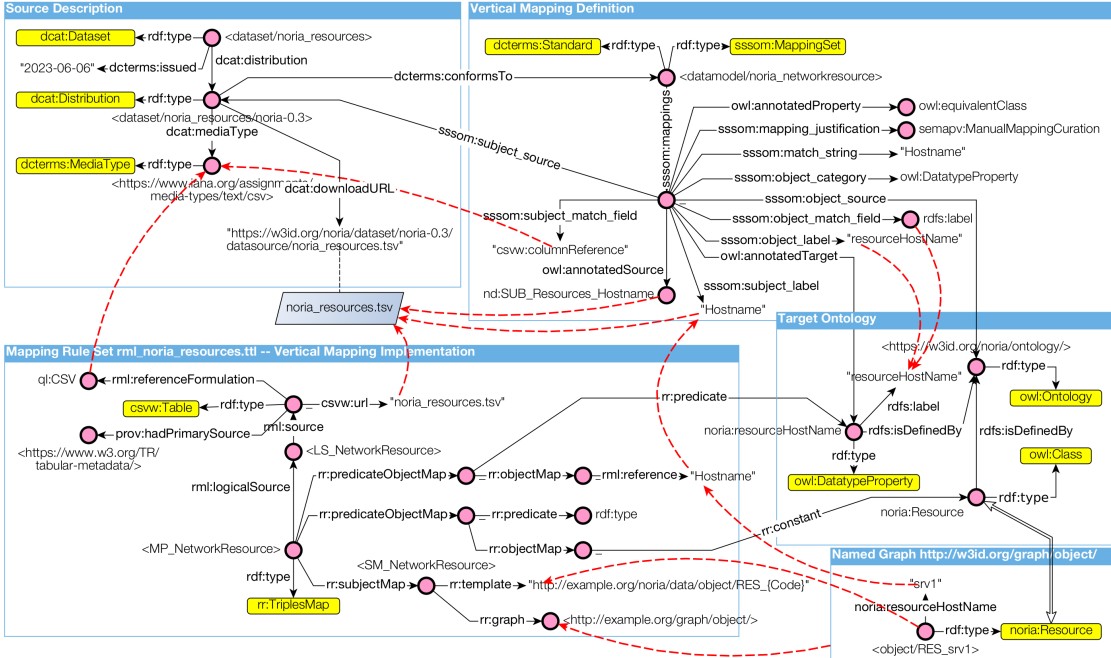

**Figure 1:** Sketching end-to-end traceability in declarative data integration.
An attempt to connect the subgraphs involved in a KGC pipeline to understand how different frameworks (RML [1], SSSOM [18], DCAT [19], the NORIA-O [20] target ontology, etc.) interact, enable semantic mapping or data lineage calculations, and what connections might be missing (dashed lines) for end-to-end traceability and tackling the "semantic gap" issue. Five subgraphs are identified: data source description (top left), conceptual mapping (top right), mapping rules (bottom left), target ontology (right), and the resulting data graph (bottom right). Paths between subgraphs suggest alignment with the framework for understanding from Figure 2, such as `noria_resources.tsv` ← [Mapping Rule Set] → `srv1` for the "data lineage" case, and (Mapping Rule Set) ← [Source Description × Ontology] → (Vertical Mapping Definition) for the "Mapping templates" case. "Conceptual mapping" refers to high-level, implementation-agnostic correspondences between source schemas and target ontology concepts, as opposed to mapping rules (e.g., RML TriplesMap), which are their executable counterpart.

are approached through theoretical and methodological proposals, fundamentally relying on mathematical formalisms (e.g., algebraic [9, 10, 11], functional [12], category theory [13]) or practitioner interviews [14]. These works are instrumental in the design and deployment of KG-based systems and applications, particularly the mathematical approaches mentioned, which bring a model-based design and testing perspective to the field.

Nevertheless, when knowledge graphs are conceptualized as *data products* [15], through the combined lenses of semantic integration and data governance, it becomes evident that maintaining semantic continuity across the full KGC pipeline remains an unsolved challenge. Essential specification-level features[1] are still insufficiently defined, hindering a principled and reproducible approach to end-to-end KGC. For instance, in Figure 1, the lack of clear connection between data source characteristics and a set of RML rules disrupt provenance calculus [16, 17]. Throughout this paper, we use "data lineage" to denote the formal record of how each piece of information in the knowledge graph originates from and is transformed by upstream source data and mapping operations, distinct from data provenance, which typically refers to the origin

and custody of data rather than its full transformation trace.

Going a step further, in a *multi-domain KGC pipeline* context [21, 22], we observe that there is no standard way to express transformations made to the data at the knowledge graph level (a.k.a. patching queries [21]), nor how these relate to the upstream RML rules [23, 24]. To illustrate the implicit challenges of this, let us consider the case of building an IT Network Digital Twin (NDT) [25]: it involves integrating diverse network segments data (e.g., IP backbone, data centers, optical networks), each requiring different graph construction approaches (i.e., ways to map data elements and link them together) aligned with specific ontologies and domains. Despite a unified vocabulary (e.g. DevOpsInfra [26], NORIA-O [20], SEAS-PEP [27]), creating the NDT KG requires developing multiple domain-specific subgraphs and stitching them into a coherent, consistent and usable multi-domain knowledge graph for all stakeholders.

These observations converge with the Shift Left Data Manifesto [28], which highlights the challenge of maintaining semantic continuity from how data is structured (schemas) to its actual meaning in context (semantics). We refer to this as the *semantic gap* issue, and we argue that end-to-end traceability – from source data to user-level usage – is the operative mechanism for achieving this continuity and, therefore, a fundamental requirement for designing and using KG-based systems. Without it, semantic integration and knowledge sharing will remain a craftsmanship task closely tied to the knowledge/data engineer's tacit understanding of the information system they work in.

In what follows, we further develop our vision by first sketching a framework for understanding and behavior-driven design of KGC pipelines (§2), and conclude by defining a high-level roadmap for future work (§3).

## 2. End-to-end Traceability as a Building Block

Based on the above discussion, it can be understood that five categories of objects need to be analyzed, both in their nature and in their relationships, to address the semantic gap issue and the multi-domain KGC challenges (Figure 2).

To further develop this intuition, we propose the following hypothesis as the foundation of our vision (**Hyp. 1 – End-to-End Traceability**): *end-to-end traceability should be a fundamental requirement for designing a KGC pipeline, if not considering a KG-based application.*

Practically, Hyp. 1 corresponds to demonstrating the following properties for any implementation of a multi-source, multi-process KGC pipeline: (**Prop. 1 – Lineage Completeness**) all source data contributing to an RDF triple are properly traced; (**Prop. 2 – Lineage Correctness**) the lineage graph [17, 29] does not contain spurious dependencies, that is, no source attribute is recorded as contributing to a triple unless a direct mapping path can be established between them; (**Prop. 3 – Lineage Minimality**) no unnecessary dependencies are added to the lineage graph.

In direct relation to the semantic gap issue, we further hypothesize (**Hyp. 2 – End-to-End Semantic**) that *end-to-end traceability should encompass a semantically analyzable description of source data (e.g., [30, 31, 32]) and a formal description of how end users leverage the KG*

---

[1]By which we mean properties of a KGC pipeline that can be expressed and verified independently of any specific implementation.

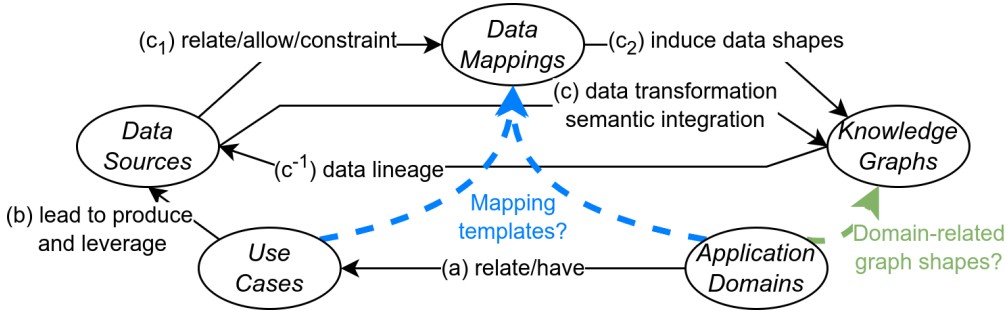

**Figure 2:** Objects involved in KGC using the category theory perspective.
With *Knowledge Graphs* as the target representation, we hypothesize that an *Application Domain* is related to ($a$ edge) a set of *Use Cases*, which in turn involve using ($b$ edge) a given set or type of *Data Sources*. These are shaped and combined through data transformation and semantic integration ($c$ edge) in a convenient manner (e.g., running an RML engine and patching queries) to meet the requirements of the application domain and use cases. Looking backward at the relationships (i.e., treating $c^{-1}$ as a modeled reciprocal of $c_2 \circ c_1 = c$, used here as an analytical device for formal verification rather than a claim of strict invertibility), we posit *Data Mappings* (e.g., RML TriplesMaps) to hold information on the use cases, representing some kind of functional dependency that enables revealing and studying *Mapping templates* and *Domain-related graph shapes* that could be generally applicable.

*data, including application domain description and implicit/explicit requirements (e.g., [33, 34]).* Practically, Hyp. 2 involves the KGC pipeline/KG-based application allowing the verification of the following two properties: (**Prop. 4 – Source Minimality**) identifying which source characteristics are mandatory for a given use case, and vice versa; (**Prop. 5 – Graph Shape**) determining the typical graph shapes [35] that can be expected from a set of sources with respect to a use case and/or an application domain. In particular, Prop. 4 directly operationalises the notion of data contracts [36]: a source attribute is mandatory if and only if it is declared as such in the contract governing the corresponding use case, making traceability not only a lineage concern but a governance one.[2]

Finally, in direct relation to the multi-domain KGC challenges and to support the scrutiny of Prop. 5, we consider that (**Hyp. 3 – Two Axes Mapping**) *the same concepts and properties of an ontology are involved in an orthogonal yet complementary manner during the KGC process, depending on whether it involves data transformation or semantic integration.* Practically, Hyp. 3 leads us to establish the following two definitions (which can be seen as lemmas for exploring the properties discussed above): **Vertical mapping** traverses abstraction levels within or across domains, connecting specific data-level representations to more general or more specialized ontological concepts, as in Figure 1 when RML rules lift tabular records into an OWL class hierarchy. **Horizontal mapping** operates across domains at equivalent levels of abstraction, establishing correspondences between conceptually similar entities in different ontological frameworks, analogous to a lateral alignment between parallel representations. This distinction impacts mapping reusability and transferability patterns, particularly relevant for understanding how declarative mapping rules can be effectively reused across different transformation contexts.[3]

---

[2]We remark that FAIR principles for KGs [37] provide an interesting additional enabling layer for Hyps. 1 & 2, as they offer the metadata infrastructure required for findable, accessible, interoperable, and reusable traceability artifacts across organizations and pipeline versions.

[3]The complementary roles of SSSOM and RML within this framework are noteworthy. SSSOM operates at the

## 3. Conclusion and Future Work

The end-to-end traceability hypothesis offers a principled basis for elevating KGC pipelines from craftsmanship-driven integration efforts to auditable data products, systems in which the relationship between source semantics, mapping logic, and graph output is formally verifiable. Systems satisfying Prop. 1 – 5 would address pressing operational needs: trustworthy multi-domain data lineage, reduced data collection footprints through source minimality, mapping rule inference grounded in explicit source and use-case descriptions rather than tacit expert knowledge.

Achieving these major objectives obviously requires experiments and intermediate implementations to further develop and test our vision. A first promising approach is to develop a *mathematical framework* that formalizes the usage context and explores data alignment via structure and semantic compatibility. For example, as suggested by [9], [11] and [38], datasets can be mathematically modeled and subjected to compatibility operations, enabling the algebraic analysis of data models. Closer to the KG realm, another option could be to address data alignment by comparing provenance information to then associate data [39, 40], whether it involves serial or parallel provenance (i.e., the chaining or parallelization of data integration traces). However, various levels of abstraction are to be explored – algebraic [9, 10, 11], functional [12], and category theory [13, 38][4] – to identify the appropriate level of generalization.

A second promising approach is to challenge multi-domain KGC at design time using *formal proofs of systems* [41]. In an RML/SPARQL context, this involves validating data lineage properties by formalizing RML operations (initial generation of RDF triples), SPARQL operations (updates to the graph with patching queries), and the update rules for the lineage graph associated with these operations. Assuming that a mathematical framework has been chosen based on the scope of properties to be studied and the expected level of generalization, the challenge is not so much to prove the theoretical framework itself but to produce a faithful translation of the RML rules and SPARQL queries into this framework, and then to seek a proof of the properties for concrete instances of these translations.

A final promising approach is to explore *relationships with non-KG or legacy technologies.* For instance, the `datacontract-cli` tool from the ODCS project [36] offers extensive SQL tooling and a strong ecosystem, but focuses more on data access than semantics, unlike LinkML [18], which aligns with metadata registry concepts and includes model alignment tools. Translating structured representations from ontologies like DPROD [42] and LinkML-SSSOM [18] into the `datacontract-cli` format could facilitate leveraging its connectors.

---

specification level, expressing semantic correspondences between concepts in a human-readable, ontology-agnostic format. In contrast, RML operates at the implementation level, encoding executable transformation logic. A specification should be substantially simpler than its implementation. In practice, an SSSOM mapping set for a given domain involves far fewer constructs than the corresponding RML TriplesMap, which must additionally encode source access patterns, value transformations, and named graph assignments. Hyp. 3 formalizes this distinction as the vertical/horizontal mapping axes.

[4]Since category theory enables modeling as much structure as the data has [13], an interesting avenue would be to explore how RML relates to category theory, considering RML as a model morphism [38] with an implicit product operator (i.e., composing data elements).

## Acknowledgments

We would like to thank Béatrice Rouxel, Yoan Chabot, Ivan Bedini, and Didier Bringer from Orange for their advice on the initial version of this project. We also thank Tobias Schweizer for early discussions on model crosswalks in relation to the MSCR project, and Vladimir Alexiev, Anelia Kurteva, and Ioannis Dasoulas (KGCW 2026 Program Committee members) for their appreciation and advice.

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
