# OpenReview forum: "Shifting Models Left: End-to-End Traceability as a Foundation for Knowledge Graphs as Data Products"
_eswc-conferences.org/ESWC/2026/Workshop/KGCW — KGCW 2026_

### Official Review · ~Vladimir_Alexiev1 · 2026-03-16
**good paper but thin on details**

**Rating:** 7
**Confidence:** 4

**Review:**

This is a good **position** paper.
I wasn't familiar with the "shifting data left" manifesto, it's a good read.
The paper proposes to operationalize this concept for KGs and calls that "shifting semantics left".
I like the first diagram that shows dataset description (DCAT), an ontology, mapping spec (SSOML), mapping implementation (RML), the resulting triples: and the missing links that could be added between them to facilitate traceability, provenance, derivation, automation, mapping modularity/reuse.

Where the paper falls short is on details:
- It would be nice to show the implementation and benefits of such linking triples
- Comment on the relative strengths and weaknesses of SSOML vs RML, and their suitability for the different KG construction steps. A spec should be an order of magnitude simpler than an implementation: is that the case here?
- The "Properties" and "Hypotheses" in sec 2 are not sufficiently motivated/elaborated, and the relations between them are not sufficiently clear. But I think that can be expected of a position paper, especially if it has a firm limit on number of pages.

Please study some of the work at https://categoricaldata.net/papers, that uses category theory to both describe datasets and the transformations between them. In particular "Algebraic Property Graphs" is joint work with Uber, and it's claimed to have facilitated the harmonization of hundreds of thousands of datasets at Uber. And add some references and discussion/comparison.

---

### Official Review · ~Anelia_Kurteva1 · 2026-03-30
**good paper addressing a rising need for better KG governance**

**Rating:** 7
**Confidence:** 4

**Review:**

The paper highlights a shift in how Knowledge Graphs (KGs) are built and managed. This is a very pressing topic requiring more attention.
I agree with the authors' view that KGs can be seen as data (or even knowledge) products and as such they should be better managed on semantic, data and use case levels. Considering the rising use of generative AI to automate many of the KGs construction tasks, this might become even more challenging than it currently is.

The hypotheses in Section 2 are valid and sound. The paper is missing a discussion on the role of FAIR KG documentation, which can assist in all these hypotheses. It would have been interesting to also include a discussion or hypothesis on how one can implement end-to-end traceability for multimodal knowledge graphs.

Overall, this is a good paper that will create a much needed discussion on the topic.

---

### Official Review · ~Ioannis_Dasoulas1 · 2026-03-30
**Interesting vision paper for facilitating semantics provenance in knowledge graph construction pipelines**

**Rating:** 6
**Confidence:** 3

**Review:**

This vision paper proposes an approach for facilitating semantic traceability within the data integration process for multiple data sources during knowledge graph construction. The method focuses on end-to-end traceability of semantics within the different frameworks that are used in knowledge graph construction.

Strengths:
1. The paper tackles a recurrent problem in data engineering pipelines that typically requires a lot of manual labor
2. The paper provides an interesting abstraction of the frameworks participating in knowledge graph construction, as well as their connections. A generalization and formalization of these frameworks seems useful for facilitating knowledge graph construction pipelines.

Weaknesses:
1. Problem Formulation: The problem formulation and motivation were a bit hard to understand initially. Some terms such as ‘conceptual mappings’ or ‘specification-level features’ are mentioned but not introduced. Figure 1 is also complex in many places hindering the understanding of the examples.
2. Parts of the method are not sufficiently motivated: The paper proposes a set of hypotheses and proposals, but these are only shortly explained. In addition, they build on a set of foundational objects involved in knowledge graph construction, also proposed in this paper. Hence, the hypotheses build on more hypotheses which do not seem to be peer-reviewed or reused from another paper. For that reason, I believe that the paper would benefit from reusing definitions and standards introduced in past works to propose new hypotheses, e.g. reuse the definition of objects involved in knowledge graph construction from past KG-specific or broader surveys, as well as relevant definitions and proposals from the provenance domain, as mentioned in the conclusion.

---

### Decision · Program_Chairs · 2026-04-09

**Decision:**

Accept

**Comment:**

This paper has been selected for presentation at the KGC workshop. We strongly encourage the authors to consider the reviews whilst revising the paper. Camera-ready instructions will soon follow.